# End-to-End Database Software Security

**Denis Ulybyshev \*, Michael Rogers, Vadim Kholodilo and Bradley Northern**

Department of Computer Science, Tennessee Technological University, Cookeville, TN 38505, USA
\* Correspondence: dulybyshev@tntech.edu; Tel.: +1-931-372-6127

**Abstract:** End-to-end security is essential for relational database software. Most database management software provide data protection at the server side and in transit, but data are no longer protected once they arrive at the client software. In this paper, we present a methodology that, in addition to server-side security, protects data in transit and at rest on the application client side. Our solution enables flexible attribute-based and role-based access control, such that, for a given role or user with a given set of attributes, access can be granted to a relation, a column, or even to a particular data cell of the relation, depending on the data content. Our attribute-based access control model considers the client's attributes, such as versions of the operating system and the web browser, as well as type of the client's device. The solution supports decentralized data access and peer-to-peer data sharing in the form of an encrypted and digitally signed spreadsheet container that stores data retrieved by SQL queries from a database, along with data privileges. For extra security, keys for data encryption and decryption are generated on the fly. We show that our solution is successfully integrated with the PostgreSQL® database management system and enables more flexible access control for added security.

**Keywords:** software security; database security; access control; data privacy

## 1. Introduction

Database management software is widely used in private and public sectors, including government, manufacturing, public utilities, e-commerce, and other domains where storage and fast retrieval of data are desired. The relational database model is widely used due to its flexibility and scalability, and its wide application to many kinds of data. Furthermore, it is easy to understand and query and has a flexible and popular language interface called Structured Query Language (SQL).

When the first relational database management software applications came into wide use, they had very little security. As their popularity grew, so did malicious attacks on relational databases in order to steal information. Therefore, more security was integrated into relational database management software over the years, both for data in storage and data delivered to the client over the network.

Unfortunately, typically, once data arrive at the client application, data are no longer secure. They are stored in files or presented in report format without encryption or access control and can be viewed by anyone that has access to the client's computer. Likewise, if a client shares these data by transferring them via email or other methods over the commodity Internet, they can be intercepted and read by unauthorized parties. A gap exists in current technology for providing server-enforced security *after* the data reach the client. In other words, the data should reach the client in a secure form that guarantees that the data remain confidential except for those that have the right to access them.

An important component of this existing gap is a lack of flexible access control both while the data are at the server and after the data reach the client. In most relational database management systems (RDBMS), access privileges can be granted to the relation (table) or to a particular column (attribute) of the table. Supporting more fine-grained access control policies by granting privileges to a particular data cell, depending on the data content

stored in that cell, is highly desirable. For instance, a given role, e.g., "gastroenterologist", can access the attribute "diagnosis", if the diagnosis begins with "gas". Our solution provides fine-grained access control and supports data integrity and secrecy after the data reach the client.

To summarize, in this paper, we propose a methodology to solve the following problems:

1. Protect database records on the relational database client side, as well as in transit.
2. Provide fine-grained role-based and attribute-based access control for database records on the client side after it leaves the server. In our access control model, for a given role or user with a given set of attributes, access can be granted to a relation, a column, or even to a particular data cell of the relation, depending on the data content.
3. Enable decentralized data access and peer-to-peer data exchanges between clients, which eliminate the necessity to contact the database server each time to request data from the database.

## 2. Motivation and Goals

Typical methods of security for most database management software are as follows. Most servers require users to log into the database server to provide authentication and authorization for access control. Once a database user is identified, the database server can enforce its rules that determine what data that user can access. A database administrator can allow users to access tables, views, or stored procedures for accessing the data. Furthermore, to control updates, a database management system (DBMS) can enforce integrity controls such as primary keys, foreign keys, type constraints, etc., that constrain the way users manipulate the data. For secrecy, database management software may encrypt files on some secondary storage and also encrypt query results for secure network delivery to the user.

Many database management software applications do not have the ability to define access controls at the column level. Instead, the method of controlling access to specific columns in tables for most database management systems is to disallow direct query access to the tables and define views and stored procedures. These views and stored procedures execute queries that may leave columns out of the result set for particular roles. Unfortunately, these methods are not as scalable for database management as role-based access control. For example, in a DBMS, the administrator has to create separate views or stored procedures for each separate user/role. For $N$ different roles, $N$ different views and/or stored procedures are needed. Moreover, changes to privileges mean that every view/stored procedure would have to be rewritten. However, if an RDBMS could support role-based access control (RBAC) at the column level, then only the access control list (ACL) would need to be changed.

Although some databases do allow column-level access control at the time of a query [1], the lack of granularity of access control is not the only issue. All of the typically supported RDBMS security measures are effective as long as the data are controlled by the server, but not after the data reach the client (user). For example, consider a scenario where a team of medical professionals needs to access the medical history report for a patient. The report is generated from an SQL query or a sequence of SQL queries. The data for that patient that are stored on the trusted server are secure, and the report is securely generated and then delivered to the head physician on a secure channel. Although the head physician should be able to read all of the report, only parts of the report may need to be read by radiologists, cardiologists, and pharmacists. However, the report should not be accessible by anyone else, and those that can access the report should only be able to read the parts for which they have authorization. In this scenario, the head physician cannot simply deliver the report to those that need to read it. If the report is delivered over an insecure network (e.g., via http), *anyone* sniffing the network can also see it. Additionally, the head physician cannot guarantee that the intended recipients will only read the parts for which they are authorized. This scenario is not atypical and can be applied to corporations,

academic institutions, and many other domains that have reports generated from relational databases that must be viewed by various departments or organizational units.

The goal of this paper is to describe a design and an implementation of PROtected SPrEadsheet container Generator for SQL (PROSPEGQL), which is a novel add-on to relational databases that supports scenarios similar to the above. In particular, this research paper has developed and integrated a container-based solution into an RDBMS that provides the following:

- *Persistent security for SQL results*. Query results are secure even after landing on the client's computer and being shared with other clients.
- *Fine-grained access control*. The container-based solution provides role-based and attribute-based access control so that only authorized entities are allowed to see the data. These ACLs are embedded into the container, so access control can be enforced no matter where the container is stored, or how it is transferred from one destination to another. Access control operates at the relation (table), attribute (column), and cell (data item) levels, depending on the access control needed.
- *Decentralized data access to the database*. A user does not need to contact the database server each time they need the data. Therefore, a single point of failure is eliminated, and query and retrieval cost is reduced.
- *Integration with any RDBMS* with stored procedure support including databases that already have cryptographic support such as PostgreSQL®, Microsoft® SQL Server® (the paper "Persistent Security and Flexible Access Control for RDBMS" is an independent publication and is neither affiliated with, nor authorized, sponsored, or approved by Microsoft Corporation® [2]) or CryptDB [3].
- *On-the-fly encryption/decryption key generation* for scalable and secure key management.

In the access control model used for our PROSPEGQL, access privileges to a particular cell (data item) are specified with regular expressions that enable great flexibility in defining access conditions. Access control follows the principle of least privilege for column data, aggregates, and computed/sorted values. In other words, the ACLs are constructed such that the user only has the privilege for the computed values that is provided by the value for which the user/role has the least privilege (see Section 4 for a further description).

The impact of our proposed solution, which can easily be deployed at commodity RDBMS servers, is that it will protect database records on the client side after data leave the database server and supply a secure way to share information among database users in a decentralized peer-to-peer way. PROSPEGQL can be used in hospital information systems, large and small organizations, and anywhere role-based access control for relational data is necessary.

### 3. Related Work

To ensure data privacy on untrusted servers, a database must store the data in encrypted form. SQL queries over encrypted data must be supported, along with a fine-grained access control. Database clients might need to access the encrypted relations (tables) or separate data attributes (columns).

A PostgreSQL® RDBMS enables different encryption options [4] to protect data in transit and at rest. PostgreSQL® supports encrypting a specific attribute in a table or encrypting the entire data partition. It also supports client-side encryption, which can be used when a database administrator is untrusted. However, the key distribution problem arises if the data owner that has encrypted the data with a key wants multiple entities to access data subsets.

The approach in [5] implemented a trust model that allowed operations in a decentralized setup but did not address access control. In contrast, our approach provides data protection in transit and at rest on a client's side and enables a fine-grained access control. This allows different authorized roles or users to access a table, a separate column, or a separate data item; these data have been encrypted by a data owner. After SQL query results land on a client's side computer, the client can share the data in the form of a

PROtected SPrEadsheet Container for DataBases (PROSPECDB) with the other parties without the necessity to communicate with a database server.

The CryptDB [3] database engine stores data on the database server in encrypted form and protects the data from untrusted database administrators. CryptDB supports a subset of SQL queries to work on encrypted data. When a client issues an SQL query, data decryption takes place on a trusted proxy server and then decrypted SQL results are sent to a client. The database server does not have access to the encryption and decryption keys [3]. If the server is compromised, then only ciphertext is revealed and data leakage is limited to data for users who are currently logged in the database. In the PROSPEGQL solution, encryption and decryption keys are *not stored* on either the client's or server's side and not on any trusted third party. The encryption keys are generated on the fly when PROSPECDB is generated. Decryption keys are generated on the fly when data in the PROSPECDB are viewed by an authorized client [6].

For supporting inequality and range queries, CryptDB supports order-preserving encryption (OPE), which is prone to frequency analysis attacks, and deterministic encryption (DET). In 2015, Naveed et al. demonstrated in [7] successful attacks on CryptDB to recover the plaintext from database columns, encrypted with DET and OPE encryption schemes. Raluca Popa in [8] presented the guidelines on how to use their CryptDB system to prevent sensitive data leakage. The DET scheme provides strong encryption guarantees only if there are no data repetitions in DET-encrypted attributes and every value is unique. OPE should not be used for columns with sensitive data. One of the solutions to replace OPE could be using a fully homomorphic encryption, but it imposes a very significant performance overhead. As an alternative, partially homomorphic encryption can be employed. A significant difference in our implementation of PROSPEGQL and CryptDB's implementation of sensitive and nonsensitive columns is that PROSPEGQL protects from inference attacks for which the OPE encryption scheme is vulnerable.

PROSPEGQL derives ACLs such that a user/role only has access to a particular column in a query result set according to the column used in any aggregate, function, expression, or ordering that has the least privilege (see Section 4 for further explanation). This scheme prevents access to computed columns and sorted results if the client does not have access rights to the columns involved in the computation/sorting.

In a PROSPECD data container, presented in [6], the smallest granularity unit for access control is a data worksheet (data subset in the spreadsheet file). In this paper, the containerization solution is extended to the PROSPECDB container to support access control not only for separate data worksheets but also for separate data columns inside the worksheets and separate data cells depending on the data content, using Perl® regular expressions [9]. It allows us to grant or deny access for a given database user or role. This access is delineated to a separate data item based on the content stored in that data item. For this paper, we created PROSPEGQL by integrating the PROSPECDB container with a PostgreSQL® RDBMS.

The conceptual difference between the PROSPECDB container and an active bundle [10–13] is that an active bundle incorporates data, metadata, and a policy enforcement engine (virtual machine), whereas PROSPECDB only stores data and metadata, without a policy enforcement engine. Furthermore, in contrast with active-bundle and P2D2-based [14] solutions, PROSPECDB detects several types of data leakages that can be made by malicious insiders to unauthorized entities [6]. Moreover, in PROSPECDB, the access control policies can be specified in the form of Perl® regular expressions that decide based on the data content whether the access privilege can be granted.

A solution by Tun and Mya in [15] encrypted the selected cells in a spreadsheet file and embedded the hash value of the content. In PROSPEGQL, each separate worksheet is encrypted with a separate symmetric key, generated on the fly, for fine-grained access control. Furthermore, access can be granted to a separate worksheet, attribute (column), or data cell depending on data content.

Almarwani et al. [16] proposed a solution that supported queries over encrypted data and a fine-grained access control. The static model was based on ciphertext-policy attribute-based encryption and CryptDB [3]. Their dynamic model was a combination of CryptDB and PIRRATE [17] that was built on attribute-based encryption and supported revoking access from users via a proxy [17]. Encrypted files could be shared in a social network and decrypted by multiple users on their side, using the proxy key. PROSPECDB differs in that it stores access control policies in encrypted form together with the encrypted data, as embedded worksheets. The PROSPECDB container is generated on a trusted server as an encrypted and digitally signed spreadsheet file, where each data worksheet stores the results of an SQL query which is encrypted with its own key, generated on the fly.

## 4. Core Design

The core design of PROSPEGQL integrates the RDBMS with a PROSPECDB container generator and a viewer, as shown in Figure 1. The client submits an SQL query through the query interface, which is processed by the integration components so that the PROSPECDB container can be generated. Finally, the container can be downloaded by an authorized client and viewed by the clients with appropriate access permissions. The following subsections describe the data flow of the system in details, followed by the design of the PROSPECDB container and an access control model. We successfully integrated our PROSPECDB container with a PostgreSQL® RDBMS using stored procedures. Furthermore, the deployment can be easily ported over to Oracle® SQL or other RDBMS products with cryptographic functionalities.

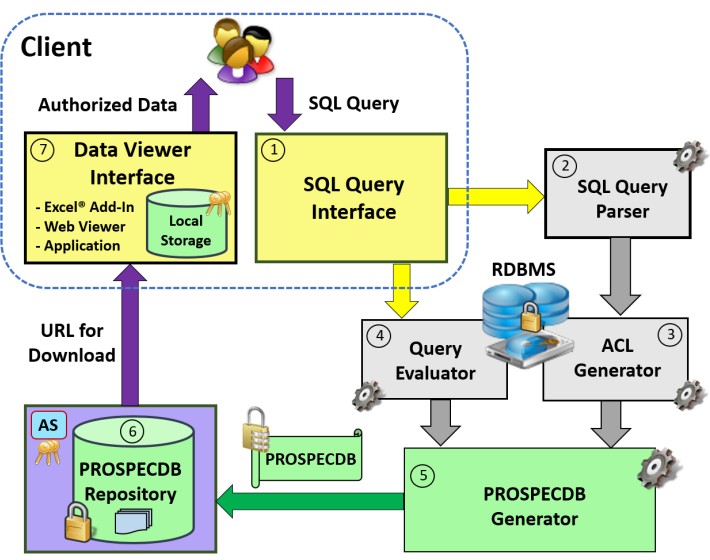

**Figure 1.** PROSPEGQL Workflow.

### 4.1. Data Flow Design

As shown in Figure 1, a client submits a query through a commodity user interface such as a terminal emulator or graphical SQL client, or via a database API of a programming language. The query constructed must call one of two PROSPEGQL functions, which are *get_container()* or *get_container_url()*, and pass the SQL query to that function as an argument. An example of a PROSPEGQL query can be found in Listing 1.

**Listing 1.** Example of PROSPEGQL get_container Query

```
SELECT get_container (
  'SELECT * FROM Patient NATURAL JOIN Billing_Info
   WHERE Patient.id = ''PB0023S''')
```

The above query returns a PROSPECDB with the SQL query results and access control privileges in encrypted form, as a binary large object (BLOB). The client can then view the BLOB using the data viewer interface, store it for later use, or even send it over the network to other parties to view and be confident that the data in the container are protected from unauthorized accesses. The *get_container_url()* function is the same as *get_container()* except that instead of returning the PROSPECDB as a BLOB to the client, it stores the PROSPECDB in the repository and returns a uniform resource loader (URL). The utility of *get_container_url()* is that the PROSPECDB is immediately available for viewing for authorized parties via the secure web-based viewer without having to download it. To generate and view the PROSPECDB data, the PROSPEGQL functions accomplish the following steps, shown in Figure 1:

1. The client issues the SQL query to the database.
2. PROSPEGQL parses the SQL query argument to determine its database objects, which include columns used in its SELECT clause and tables in its FROM clause.
3. The function generates an access control list (ACL) by querying the database server for database privileges for the discovered columns and tables.
4. PROSPEGQL then evaluates and executes the SQL query on the database server to obtain a result set.
5. PROSPEGQL passes the result set and an ACL to the PROSPECDB container generator, which then generates the container.
6. PROSPEGQL then stores the resulting container in the PROSPECDB repository or, if the function is get_container() instead of get_container_url(), it passes the container back to the caller as a BLOB.
7. An authorized client can view PROSPECDB data in a Microsoft® Excel® Add-in, a standalone application, or in a web viewer. The authorized client communicates with the authentication server (AS) to derive decryption keys for accessible PROSPECDB data subsets. Details are described in Section 4.2.

The construction of the ACLs in step 3 is accomplished in such a way as to reduce the threat of inference attacks. For example, a client submits the following query in Listing 2.

**Listing 2.** Example Query for Reduction of Inference Attacks

```
SELECT base_treatment_rate * sales_tax_rate
FROM Billing_Info
```

Consider a particular role that has read privileges for `sales_tax_rate` but not for `base_treatment_rate`. Step 3 would create an ACL for the column in the "Results" sheet for the expression "`base_treatment_rate * sales_tax_rate`". The privilege for that particular role would be the same as that role's privilege for the `base_treatment_rate` column in the database. In other words, the role would not be able to read the "`base_treatment_rate * sales_tax_rate`" column in the result set. Furthermore, consider the query in Listing 3.

**Listing 3.** Example Query for Demonstrating Roles

```
SELECT name, base_treatment_rate
FROM Billing_Info
ORDER BY base_treatment_rate
```

Again, consider that a particular role does not have read access to `base_treatment_rate`. A member of that role hopes to infer the values of the database column from the column in the result set according to its sorted order. However, step 3 will consider an ORDER BY clause to be an expression over the entire query (i.e., all the result set's columns). Therefore, the ACLs for the columns in the query result set will be constructed such that each will have the lowest privilege level of all the columns used in its expression, including the columns in the ORDER BY clause. In this case, the role will not have the privileges to read any of the columns in the result set, and thus members of that role will be able to infer nothing.

*4.2. PROSPECDB Data Container*

A PROtected SPrEadsheet Container for DataBases (PROSPECDB) is a spreadsheet file that stores data subsets as separate encrypted data worksheets, along with an encrypted "Metadata" worksheet. Data worksheets are encrypted with separate symmetric keys that are generated on the fly. A "Metadata" worksheet contains access control policies encrypted with a separate predefined key. The on-the-fly key generation procedure takes the following inputs:

1. The hash value of the authentication server's (AS) private key. A data viewer sends an https POST request with the attached X.509 certificate of the client to the AS, which verifies the client's identity. The AS can only send the hash value of its private key to authorized clients, based on their roles. This hash value can be cached locally. The AS manages access revocations.
2. The hash value of metadata, which contains access control policies. Including this component in the symmetric key generation protects the PROSPECDB from unauthorized modifications of metadata.
3. The hash value of the worksheet's name, i.e., the data subset name.

The SHA-256 hash function is used. The on-the-fly symmetric key generation procedure is the same as that used in a PROSPECD container [6]. For encryption in the PROSPECDB, the "CryptoJS" library written in native JavaScript [18] is used. The new PROSPECDB feature introduced in this paper compared to PROSPECD is the access that can be granted to a separate attribute (data column in a worksheet) or a separate data cell (data item), depending on the content of this data item. The PROSPECDB is mapped to the relational model: columns in the container are attributes in the relational model. PROSPEGQL technology integrates the PROSPECDB into an RDBMS by incorporating both database ACLs and SQL query results into the encrypted worksheets. For instance, an electronic health record of a patient can be created as a result of querying multiple tables that contain clinical and administrative information.

Data, encrypted in a PROSPECDB, can be accessed and viewed in one of these three options:

1. A cross-platform application installed on the client's side.
2. A Microsoft® Excel® Add-in, installed on the client's side.
3. A remote web viewer that runs on the same node as the PROSPECDB repository—see Figure 1.

To view the data, a client needs to enter credentials in the viewer. Based on the entered credentials, the client's role is determined and data subsets available for this role are decrypted and displayed. The viewer enforces the access control policies, either on the client's side or on a remote trusted server, depending on which of the above three options the client selected to view the PROSPECDB data.

4.2.1. Adversary Model

PROSPEGQL, integrated with PostgreSQL®, protects from the following type of adversaries:

1. *A malicious database administrator who tries to view data on the database server*. To protect data, a client must encrypt sensitive database table(s) or separate columns with their own encryption key using the native encryption support of the database server. PostgreSQL® supports several encryption modes [4]. Decryption keys are not stored on the server side. Our PROSPEGQL solution is RDBMS-agnostic and instead of PostgreSQL®, other RDBMS supporting the client-side encryption can be used.
2. *A client tries to gain access to a PROSPECDB data subset for which the client is not authorized*. Because the hash value of the "Metadata" worksheet that includes ACLs (see Table 1) is one of the inputs for the decryption key generation [6], a modification of the access control policies would lead to the wrong Advanced Encryption Standard (AES) [19] decryption key generation. Inaccessible data worksheets are encrypted

with the AES protocol, using the cipher block chaining (CBC) mode and a 256-bit key, and hidden from unauthorized clients in the viewer or Microsoft® Excel® Add-in. Breaking the 256-bit AES encryption scheme is computationally infeasible. Based on our assumptions, listed below, software to decrypt and view PROSPECDB containers (PROSPECDB viewer) is trusted. PROSPECDB viewer software is digitally signed by trusted authority to guarantee its integrity and authenticity.

3. *An adversary has access to the client's computer but does not know the client's credentials for PROSPECDB, and who tries to steal the client's data from PROSPECDB.* Similar to the item above, because the PROSPECDB spreadsheet file is encrypted, breaking the 256-bit AES encryption is computationally infeasible.

4. *An adversary tries to steal the data, sent by a client to another user, while data are in transit.* Depending on the use case, sending data in plaintext might be a violation of known policies and regulations, for example, the Health Insurance Portability and Accountability Act (HIPAA) in the healthcare domain or the Family Educational Rights and Privacy Act (FERPA) in the education domain. Thus, the client should transfer data to another user only in a protected form, such as a PROSPECDB. Since the PROSPECDB file is encrypted, even if the data communication channel is unprotected, an attacker will not be able to access the data without breaking the 256-bit AES key, which is computationally infeasible.

5. *SQL Inference attack* [20]. An authorized and malicious client tries to determine data they do not have a privilege to access by constructing an SQL expression that involves the column for which the client is not authorized. However, the client will not have the privileges to access the column and use it in any expression because of the way that the ACLs are constructed as described in Section 4.1 step 3.

**Table 1.** Access Control List with Regular Expressions.

| Columns | Admin | Doctor | Insurance | Analyst |
|---|---|---|---|---|
| MedicalInfo.ID | 1 | 1 | 0 | 0 |
| MedicalInfo.Date-of-Visit | 1 | 1 | 0 | 1 |
| MedicalInfo.Doctor's ID | 1 | 1 | 0 | 1 |
| MedicalInfo.Diagnosis | 1 | 1 | 0 | gas.* |
| MedicalInfo.Prescription | 1 | 1 | Sulfa.* | 1 |
| MedicalInfo.Blood Pressure | 1 | 1 | 0 | 1 |

### 4.2.2. Assumptions

In order to thwart the above adversaries, we designed PROSPEGQL with the following assumptions in mind:

1. Hardware and an operating system on a database server and on a client's side are trusted.

2. As long as a database server is trusted and encrypts relations or separate relation attributes (columns) stored in the server, then PROSPEGQL does not need to trust a database administrator.

3. The PROSPECDB viewer is trusted. It does not leak decryption keys and displays only the decrypted worksheets for which the client is authorized.

### 4.2.3. Security Analysis

In the probabilistic model of Goldwasser and Micali, "extracting any information about the cleartext from the ciphertext is hard on the average for an adversary with polynomially bounded computational resources" [21]. An adversary should not distinguish between the ciphertexts obtained from two plaintexts $M_0$ and $M_1$. Using the indistinguishability under chosen plaintext attack (IND-CPA) experiment, similarly to [22], and the concrete approach to define negligibility [23], we can show that the probability for an adversary to succeed in breaking the encryption scheme, used in PROSPECDB, is negligible. "A scheme is (t, $\varepsilon$)-secure if every adversary running for time at most t succeeds in breaking the scheme with

probability at most $\varepsilon$" [23]. AES with a 256-bit key may be expected to be (t, t/2$^n$)-secure as $2^{64}$ seconds are more than 584 billion years. If n = 256, t = $2^{64}$, then $\varepsilon = 2^{-192}$. A probability of $\varepsilon = 2^{-192}$ for an adversary to break the encryption is computationally infeasible.

### 4.3. Access Control Design

Our access control design consists of two major components: role-based access control and attribute-based access control.

#### 4.3.1. Role-Based Access Control.

The PROSPECDB stores an ACL in the "Metadata" worksheet that is encrypted with a symmetric AES key [6]. As shown in Table 1, the ACL defines access for roles to a given column or a given data cell. These data privileges are retrieved from the database catalog. For the current implementation of PROSPEGQL, a privilege can be *read* or *none*, represented as 1 and 0 respectively in Table 1. Further granularity for read access is specified using Perl® regular expressions by extending the database catalog with extra tables understood by PROSPEGQL. For example, the "Sulfa.*" record means that the role "insurance" can view a "Prescription" column of a "MedicalInfo" data worksheet if the data string in that column starts with "Sulfa". Thus, our model supports data access based on the data content. This extends the access control capability provided by traditional RDBMS, such as PostgreSQL®, Microsoft® SQL Server®, etc.

As explained in step (3) in Section 4.1, ACLs are generated by querying the database catalog for privileges for columns and tables identified in the SQL query. Then, these ACLs are encoded in the "Metadata" worksheet of the PROSPECDB container, as described in step (5) in Section 4.1. ACLs reference the column indices in the "Results" worksheet, not the SQL column names. Such an encoding supports SQL expressions, such as mathematical expressions, and supports access control based on data content via regular expressions. For SQL expressions on columns, the principle of least privilege is observed. In other words, the ACL is built such that, for a particular user/role, the minimum access privilege is encoded in the ACL. Likewise, the principle of least privilege is also used when building the ACLs for columns that have access rules based on data content via regular expressions. If an access rule based on data content is defined for a column used in the SQL, then the ACLs are created with the least privilege for that user/role. As described in the related work found in Section 3, this scheme protects against certain inference attacks, including attacks on OPE.

In all data access models, symmetric keys to encrypt and decrypt data are generated on the fly for a given user as defined by the role, based on three inputs, discussed in Section 4.2 and in [6].

#### 4.3.2. Attribute-Based Access Control

When a client service requests data from a PROSPECDB, the following client's attributes are evaluated: the type and version of the operating system and web browser (for data access in the web viewer), the type of the device, and an authentication method [24]. A user-agent string is retrieved from the user's client. Each user-agent type is stored in an array, which points to an attribute (bucket) in a hash table "Attribute", as illustrated in Figure 2. This hash table stores all possible names for a given attribute, such as names of web browsers or operating systems. Once the attribute name is found in the "Attribute" hash table, a second hash table, which stores version numbers and access rankings, associated with each version, is queried. To evaluate the total access ranking for the user, each attribute is evaluated by its importance and intrinsic security. The end goal is to restrict access for clients with insecure attributes. To decrease the amount of time used in searching for each attribute, we employ the use of a multithreaded model.

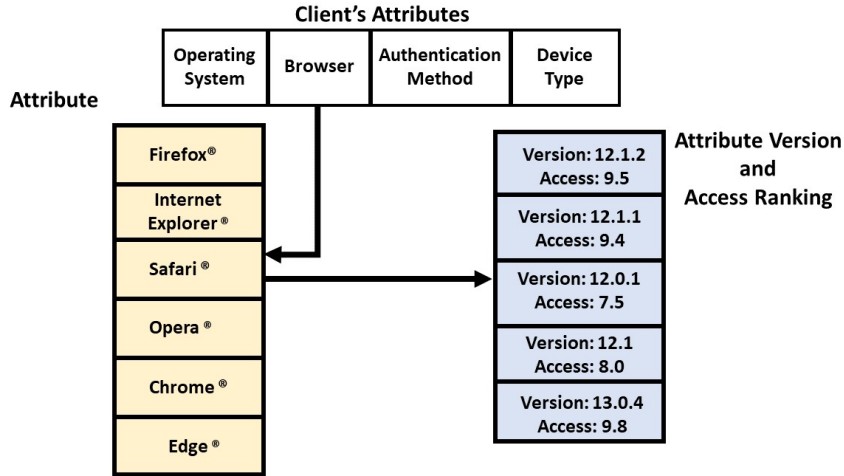

**Figure 2.** Data Structure for Attribute-based Access Control.

### 5. Evaluation

In this section, we evaluated the performance of our PROSPEGQL solution. The system configuration for our experiment was as follows:

- CPU: Intel® Core™ i5-8250U @ 1.7 GHz; RAM: 8 GB DDR4
- OS: Microsoft® Windows® 10 Pro, 64 bit
- RDBMS: PostgreSQL®, 11

Firstly, we measured time to query the PostgreSQL® database and generate the PROSPECDB that included access control policies and query results, according to steps 1–5 discussed in Section 4.1. The SQL query used in this test can be seen in Listing 4.

**Listing 4.** Test Query to Get PROSPECDB Container with Billing_info

```
SELECT get_container ( 'SELECT * FROM Billing_Info' )
```

As shown in Table 2, PROSPEGQL was considerably slower than PostgreSQL® for queries with no encryption. Fortunately, the encryption was incurred only once during the PROSPECDB generation for data that needed to be viewed by many entities without the necessity to query a database again. Next, we wanted to determine if the performance hit was because of the overhead of PROSPEGQL. Therefore, we measured the component times for PROSPEGQL. In particular, we compared the total data retrieval time from the PROSPECDB with the decryption time for data in the PROSPECDB. As seen in Table 3, columns 3 and 4, almost all the overhead was in the decryption that was imposed when viewing the PROSPECDB. Data decryption contributed from 73.3% up to 99.9%, depending on data size, to the data retrieval time.

**Table 2.** PROSPEGQL Generation Time.

| Data Size (Kbytes) | Plaintext Data Retrieval from PostgreSQL®, (ms) | Total PROSPEGQL Generation Time, (ms) |
|---|---|---|
| 0.5 | 17 | 1600 |
| 2 | 22 | 1686 |
| 8 | 46 | 1888 |
| 32 | 368 | 2296 |
| 128 | 426 | 24,654 |
| 512 | 1594 | 31,836 |
| 2048 | 6406 | 41,687 |

**Table 3.** Data Retrieval Time: Encrypted PostgreSQL® vs. PROSPECDB Decryption-only vs. PROSPECDB vs. Encrypted JSON.

| Data Size (Kbytes) | Encrypted PostgreSQL® Data Retrieval Time, (ms) | PROSPECDB Data Decryption Time, (ms) | PROSPECDB Data Retrieval Total Time, (ms) | Encrypted JSON Data Retrieval Time, (ms) |
|---|---|---|---|---|
| 0.5 | 12 | 120 | 152 | 40 |
| 2 | 22 | 135 | 160 | 53 |
| 8 | 43 | 210 | 247 | 113 |
| 32 | 106 | 520 | 614 | 246 |
| 128 | 395 | 8592 | 8893 | 946 |
| 512 | 1808 | 13,820 | 13,837 | 3780 |
| 2048 | 25,861 | 16,525 | 16,594 | 10,842 |

A client (web service) communicates with PROSPECDB container via the http protocol. To send an https GET request to the PROSPECDB for data retrieval, we used ApacheBench® (we do not claim association or endorsement of/for/by the Apache Software Foundation (ASF) [25]), version 2.3. Similarly to [6], the PROSPECDB data retrieval time started with the https GET request, sent by ApacheBench® to the PROSPECDB, and ended with the response reception. The retrieval time included times spent for authentication, access control policies evaluation, data decryption, and retrieval [6]. Data retrieval times were measured as an average of 100 data requests. The client (web service) ran on the same computer with a PostgreSQL® database, PROSPECDB, and encrypted JavaScript Object Notation (JSON) file, to exclude network delays from the time measurements. We compared the SQL query execution and retrieval times for encrypted data columns in PostgreSQL® with PROSPECDB data retrieval times, as seen in Table 3, columns 2 and 4. The following SQL query was used for decrypting and pulling data from encrypted data columns in PostgreSQL®. In the query, as shown in Listing 5, we used eight columns, with 64 bytes of data in each of them. We varied the number of encrypted records in the PostgreSQL® table to get the required data size for the experiment.

**Listing 5.** Example Query to Decrypt Columns in PostgreSQL®

```sql
SELECT encode(decrypt(
  <column1_name>::bytea, 'decrypt_key','aes_cbc'), 'escape'),
  <column2_name>::bytea, 'decrypt_key','aes_cbc'), 'escape'),
  <columnN_name>::bytea, 'decrypt_key','aes_cbc'), 'escape')
  -- Where columnN is further columns 3 to 8.
FROM <table_name>
```

Table 3 (columns 2 and 4) shows that the performance of PostgreSQL® with encryption and PROSPECDB followed a similar trend, and it corroborated our conclusion that encryption/decryption was the bulk of the PROSPECDB generation overhead. Therefore, the overhead of securing the data from the database depended upon the implementation of the encryption algorithm, and both PROSPEGQL and PostgreSQL® make use of AES. Note that the piece of hardware we ran our tests on did not support the AES-NI instructions. PROSPECDB containers support policy enforcement and data decryption on the client side, eliminating a single point of failure compared to server-based policy enforcement and decryption. For this reason, we did not assume that the client's hardware had AES-NI support. The PROSPECDB protects data on the client's side and distributes the load for decryption and policy management. Furthermore, the container relies on the on-the-fly key generation scheme, which adds an extra security layer to protect data [6]. Therefore, we believe the advantages of using PROSPEGQL are justified. Furthermore, retrieving

2048 Kbytes from PROSPECDB was 36% faster than from the encrypted PostgreSQL® table. For other data sizes, PostgreSQL® was 5.74 to 22.51 times faster.

Finally, we measured the data retrieval times for an encrypted JSON file and compared them with the PROSPECDB data retrieval times. We chose JSON for a functional reason since it is a universal data format. However, it ended up performing better than the spreadsheet files in the PROSPECDB, as it can be seen in Table 3, columns 4 and 5. We believe that the reason for the speed increase was because even though the sizes of the retrieved data in our experiments were the same, the JSON files were smaller than the spreadsheet files because of the metadata that Microsoft® Excel® includes. Furthermore, JSON files do not use decompression. Depending on the data sizes, JSON was faster than the PROSPECDB by 1.53 to 9.4 times.

To improve the performance of encryption and decryption operations for the PROSPECDB, we are working on a microservice-based implementation and investigating different cryptographic libraries. We are also investigating a C++ version of PROSPEGQL, originally implemented in JavaScript and Python.

## 6. Conclusions

Our approach improved database security by providing data protection in transit and at rest on the client's side. The developed solution was integrated with a PostgreSQL® RDBMS and extended its access control model by supporting role-based and attribute-based access control for separate data columns and data items in the relations, depending on data content. The methodology enabled a decentralized data access and peer-to-peer data exchanges between clients, which eliminated the necessity to contact the database server each time to request data from the database. Data encryption relied on an on-the-fly key generation, which made the scheme more secure since the key was not stored on the database server or inside the PROSPECDB data or on the client's side. The added functionality did add an extra overhead, compared to PostgreSQL®, depending on the data size. It was mitigated by the fact that a client had to run the SQL query only once to receive the data results that needed to be viewed by many entities.

## 7. Future Work

We plan to increase the PROSPEGQL feature support of PostgreSQL® to support the full range of Postgres® query capabilities. We also plan to support other container formats, such as Extensible Markup Language (XML), for easier integration into existing software. Additionally, we plan on optimizing PROSPEGQL by using in-memory and in-process operations instead of passing the data to and from the PROSPECDB container as files. Likewise, we will more tightly integrate the container functions into the source of the DBMS instead of the interpreted stored procedure language in which it is now written.

To address scaling for big data, we are working on a streaming solution and investigating using streaming JSON as our transfer protocol. A secure data container will be created on the client side and data from cache memory will be appended to it. Encrypted data from the database on the server side will be transferred to the client side and stored in memory, using Redis®, an open-source in-memory data store that can be used as a database cache [26]. All cached data will always remain encrypted to prevent data access if the system is attacked using one of the memory attacks, such as a *cold-boot* attack [27].

**Author Contributions:** Conceptualization, D.U. and M.R.; methodology, D.U. and M.R.; software, D.U., M.R., V.K. and B.N.; validation, V.K., D.U. and B.N.; formal analysis, D.U.; investigation, D.U., M.R. and B.N.; resources, D.U.; data curation, B.N.; writing—original draft preparation, D.U., M.R. and B.N.; writing—review and editing, D.U., M.R., V.K. and B.N.; visualization, D.U. and B.N.; supervision, D.U. and M.R.; project administration, D.U.; funding acquisition, D.U. All authors have read and agreed to the published version of the manuscript.

**Funding:** This research received no external funding.

**Institutional Review Board Statement:** Not applicable.

**Informed Consent Statement:** Not applicable.

**Data Availability Statement:** For evaluating our methodology, we used synthetic (artificially created) data to populate database tables. We did not use any publicly archived datasets or restricted datasets.

**Conflicts of Interest:** The authors declare no conflict of interest.

## Abbreviations

The following abbreviations are used in this manuscript:

| | |
|---|---|
| SQL | Structured Query Language |
| DBMS | Database management system |
| RDBMS | Relational database management system |
| RBAC | Role-based access control |
| ACL | Access control list |
| PROSPEGQL | PROtected SPrEadsheet Container Generator for SQL |
| PROSPECDB | PROtected SPrEadsheet Container for DataBases |
| URL | Uniform resource loader |
| AS | Authentication server |
| AES | Advanced Encryption Standard |
| CBC | Cipher block chaining |
| HIPAA | Health Insurance Portability and Accountability Act |
| FERPA | Family Educational Rights and Privacy Act |
| IND-CPA | Indistinguishability under chosen plaintext attack |

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
