# Peer review of "End-to-End Database Software Security"

_2674-113X, doi:10.3390/software2020007_

Round 1
Reviewer 1 Report
This paper studies the database security issues and reveals the potential threats in end-to-end database processing flow when the data is stored on the client side. The existing approaches mostly focus on the protection at the server side but ignores the risks at the client side. This paper tries to bridge this gap and introduce attribute-based access control model.
The key idea is to add a shim control layer on the client side, and provide some access authentication and container isolation. It encapsulate the raw data returned to the client, so that adverasaries' power are much limited. I like this design and believe it can strengthen the security of the DBMS.
The evaluation is also sound and the results support their claims in the paper. The related work is well written and gives me an overview of the background. Here I only have some minor issues that the authors may supplement a bit in the paper:
1. This mechanism is intergrated with PostgreSQL. I would like to see the authors' discussion on the compatibility of their mechanim and whether it can be adapted to other RDMS
2. This mechanims only targets single machine's execution, right? Because I have not seen the related security mechanims on the channels. In a distributed setting, how do you protect the channel and ensure the transmitted packets are not hijacked or forged?
All in all, a very nice paper.
Author Response
Comment 1. This mechanism is integrated with PostgreSQL. I would like to see the authors' discussion on the compatibility of their mechanism and whether it can be adapted to other RDMS.
Answer: the statement has been added to the beginning of the Core Design section - lines 215-217.
Comment 2. This mechanism only targets single machine's execution, right? Because I have not seen the related security mechanisms on the channels. In a distributed setting, how do you protect the channel and ensure the transmitted packets are not hijacked or forged?
Answer: at this time, our solution only supports single machine's execution for a database server. As a future work, we will investigate supporting federated system in a distributed setting.
Reviewer 2 Report
The paper is well written and covers a very important topic of database security. I have following concern.
1. Author should evaluate the security of proposed approach using their threat model. They should evaluate the privacy concerns as well.
2. Author should provide table of notation and clear define the problems they are trying to solve.
3. Author should provide their system setup or resources under which they have evaluated their proposed approach.
4. Author should clearly mentioned the impact of their approach and how it can be deploy in real setup.
5. Author should cite these important papers.
Privacy-preserving Crowd-sensed trust aggregation in the User-centeric Internet of People Networks ACM Transactions on Cyber-Physical Systems,2020
Author Response
(x) English very difficult to understand/incomprehensible
Answer: the updated paper draft has been carefully revised by two English native speakers, who are paper co-authors: Dr. Michael Rogers and Bradley Northern
Comment 1. Author should evaluate the security of proposed approach using their threat model. They should evaluate the privacy concerns as well.
Answer: security analysis is included in section 4.2.3. We calculated the probability for an adversary to break the encryption scheme used in PROSPECDB data container. Privacy concerns and types of adversaries are discussed in section 4.2.1.
Comment 2. Author should provide table of notation and clear define the
problems they are trying to solve.
Answer: Problem statement has been clarified in the end of the "Introduction" section, lines 45-54. Table of notations and abbreviations has been created, Table 4, line 520.
Comment 3. Author should provide their system setup or resources under
which they have evaluated their proposed approach.
Answer: system configuration under which the proposed approach has been evaluated is specified in section 5 "Evaluation", lines 417-426.
Comment 4. Author should clearly mentioned the impact of their approach and how it can be deploy in real setup.
Answer: statement on the Impact of the proposed solution has been added to the end of the section 2.1 "Motivation and Goals", lines 121-126. The way our solution can be deployed is discussed in the "Core Design" section, lines 208-217. More details are provided in lines 219-251.
5. Author should cite these important papers.
Privacy-preserving Crowd-sensed trust aggregation in the Usercenteric
Internet of People Networks ACM Transactions on Cyber-Physical Systems,2020
Answer: the suggested paper citation has been added to the "Related Work" section, lines 138-145.